# Cooperation between tropomyosin and α-actinin inhibits fimbrin association with actin filament networks in fission yeast

Jenna R Christensen[1†], Kaitlin E Homa[1†], Alisha N Morganthaler[1], Rachel R Brown[1], Cristian Suarez[1], Alyssa J Harker[2], Meghan E O'Connell[1], David R Kovar[1,2]*

[1]Department of Molecular Genetics and Cell Biology, The University of Chicago, Chicago, United States; [2]Department of Biochemistry and Molecular Biology, The University of Chicago, Chicago, United States

**Abstract** We previously discovered that competition between fission yeast actin binding proteins (ABPs) for binding F-actin facilitates their sorting to different cellular networks. Specifically, competition between endocytic actin patch ABPs fimbrin Fim1 and cofilin Adf1 enhances their activities, and prevents tropomyosin Cdc8's association with actin patches. However, these interactions do not explain how Fim1 is prevented from associating strongly with other F-actin networks such as the contractile ring. Here, we identified α-actinin Ain1, a contractile ring ABP, as another Fim1 competitor. Fim1 competes with Ain1 for association with F-actin, which is dependent upon their F-actin residence time. While Fim1 outcompetes both Ain1 and Cdc8 individually, Cdc8 enhances the F-actin bundling activity of Ain1, allowing Ain1 to generate F-actin bundles that Cdc8 can bind in the presence of Fim1. Therefore, the combination of contractile ring ABPs Ain1 and Cdc8 is capable of inhibiting Fim1's association with F-actin networks.
DOI: https://doi.org/10.7554/eLife.47279.001

*For correspondence:
drkovar@uchicago.edu

†These authors contributed equally to this work

Competing interests: The authors declare that no competing interests exist.

## Introduction

As in many cell types, the unicellular fission yeast assembles diverse actin filament (F-actin) networks within a crowded cytoplasm to facilitate different cellular functions such as cytokinesis (contractile ring), endocytosis (actin patches) and polarization (actin cables). These F-actin networks each possess a defined set of actin-binding proteins (ABPs) that regulate the formation, organization, and dynamics of the actin filaments within the network. However, the mechanisms by which specific sets of ABPs sort to particular F-actin networks are less clear. We hypothesize that a combination of competitive and cooperative interactions between different ABPs for association with F-actin is critical for establishing and maintaining their sorting. We previously identified competitive binding interactions between three fission yeast ABPs with distinct network localizations—fimbrin Fim1 and ADF/cofilin Adf1 (endocytic actin patches) and tropomyosin Cdc8 (cytokinetic contractile ring) (hereafter called Fim1, Adf1 and Cdc8)—that help facilitate their sorting to the proper F-actin networks (*Christensen et al., 2017*; *Skau and Kovar, 2010*). Specifically, synergistic activities between Fim1 and Adf1 rapidly displace Cdc8 from F-actin networks such as actin patches (*Christensen et al., 2017*; *Skau and Kovar, 2010*). However, these interactions do not explain how Fim1 is prevented from strongly associating with other F-actin networks such as the contractile ring. Therefore, we sought to determine whether other ABPs at the contractile ring prevent Fim1 association. In this study, we demonstrate that Fim1 competes with the contractile ring ABP α-actinin Ain1 (hereafter called Ain1) for association with F-actin, and that their ability to compete is dependent upon their residence time on F-actin. Additionally, we show that although Fim1 outcompetes both Cdc8 and

Ain1 individually, Cdc8 and Ain1 have synergistic effects. Cdc8 enhances Ain1-mediated F-actin bundling, and Ain1 facilitates association of Cdc8 with Fim1-bound F-actin. These combined effects allow the combination of Cdc8 and Ain1 to successfully compete with Fim1 for association with F-actin.

## Results

### F-actin crosslinking proteins fimbrin Fim1 and α-actinin Ain1 compete at the contractile ring and at actin patches

We previously found that fimbrin Fim1 and ADF/cofilin Adf1 synergize to displace tropomyosin Cdc8 from F-actin (*Christensen et al., 2017*), which helps explain why Fim1 is highly concentrated on actin patches, whereas Cdc8 is not (*Skau and Kovar, 2010*). Conversely, despite Fim1 being a dominant competitor for F-actin (*Christensen et al., 2017*; *Skau and Kovar, 2010*), Fim1 is present at much lower concentrations at the contractile ring, an F-actin network where Cdc8 is abundant (*Arai et al., 1998*; *Nakano et al., 2001*; *Wu et al., 2001*). We hypothesized that competition with additional contractile ring ABPs may prevent Fim1 from strongly associating with the contractile ring. As Fim1 is highly concentrated in actin patches, we speculated that depletion of actin patches by the Arp2/3 complex inhibitor CK-666 (*Burke et al., 2014*; *Nolen et al., 2009*) would result in a rapid increase of free Fim1 in the cytoplasm that might subsequently allow Fim1 to outcompete its contractile ring ABP competitors. Upon treating fission yeast cells expressing the general F-actin marker Lifeact-GFP with CK-666, we observed a depletion of actin patches and the formation of excessive formin-mediated 'ectopic' actin cables and contractile ring material (*Figure 1A*) (*Burke et al., 2014*).

In control (DMSO-treated) fission yeast cells, Fim1-GFP localizes predominantly to actin patches, with only a small amount associating with the contractile ring (*Figure 1B*, left) (*Wu et al., 2001*). However, in cells treated with CK-666, Fim1-GFP strongly associates with the contractile ring and to a subset of ectopic F-actin (*Figure 1B*, right). The localization of most contractile ring ABPs, including formin Cdc12, type II myosin Myo2, myosin regulatory light chain Rlc1, the IQGAP Rng2, and tropomyosin Cdc8, is unaffected by CK-666 treatment, (*Figure 1—figure supplement 1* and *Figure 1—figure supplement 2*). Conversely, less α-actinin Ain1 associates with the contractile ring in cells treated with CK-666 (*Figure 1C*, *Figure 1—figure supplement 1*). Therefore, we hypothesized that Fim1 and Ain1 are competitors, and that enhanced Fim1 association with the contractile ring in cells treated with CK-666 displaces Ain1. We tested this hypothesis by observing Ain1 localization in a strain lacking Fim1 (*fim1-1Δ*, Ain1-GFP). In *fim1-1Δ* cells, similar amounts of Ain1-GFP are associated with the contractile ring in control and CK-666-treated cells (*Figure 1D*), suggesting that the absence of competitor Fim1 allows Ain1 to remain associated with the contractile ring in the presence of CK-666. At all stages of contractile ring assembly and constriction, Fim1 similarly localizes to the contractile ring and displaces Ain1 following CK-666 treatment (*Figure 1—figure supplement 3*), although it is most prominent in stages with fully-developed contractile rings (stages 2 and 3).

If competition between Fim1 and Ain1 is a primary driver of their sorting to distinct F-actin networks, we expected that Ain1-GFP might erroneously localize to actin patches in the absence of Fim1. However, Ain1-GFP is observed at actin patches in less than 1% of *fim1-1Δ* cells (*Figures 1D* and *2C*, *Figure 2—video 1*). It is possible that a combination of the low number of Ain1 molecules (~3,600 ± 500 [*Wu and Pollard, 2005*]), and the high density of F-actin in actin patches (5,000–7,000 actin molecules in each of 30–50 actin patches [*Sirotkin et al., 2010*; *Wu and Pollard, 2005*]), may dilute the Ain1-GFP signal beyond detection. Therefore, increasing the concentration of Ain1-GFP could allow observable Ain1-GFP at actin patches, but only in a *fim1-1Δ* background. To increase the expression of Ain1-GFP, we introduced an additional copy of Ain1-GFP at the leu1-32 locus under the medium-strength 41Xnmt promoter (*Li et al., 2016*). We first quantified the cellular expression of endogenously tagged Ain1-GFP and the overexpressed Ain1-GFP constructs used in this study. We observed that while most Ain1-GFP constructs are expressed similarly to endogenously-tagged Ain1, the amount of Ain1-GFP overexpressed under the 41Xnmt promoter is almost two-fold higher (*Figure 2—figure supplement 1*). As predicted, despite this two-fold increase in expression, Ain1-GFP localizes to actin patches in only 10% of WT cells expressing endogenous Fim1 (*Figure 2A,C*, *Figure 2—video 1*). In contrast, overexpressed Ain1-GFP associates with actin

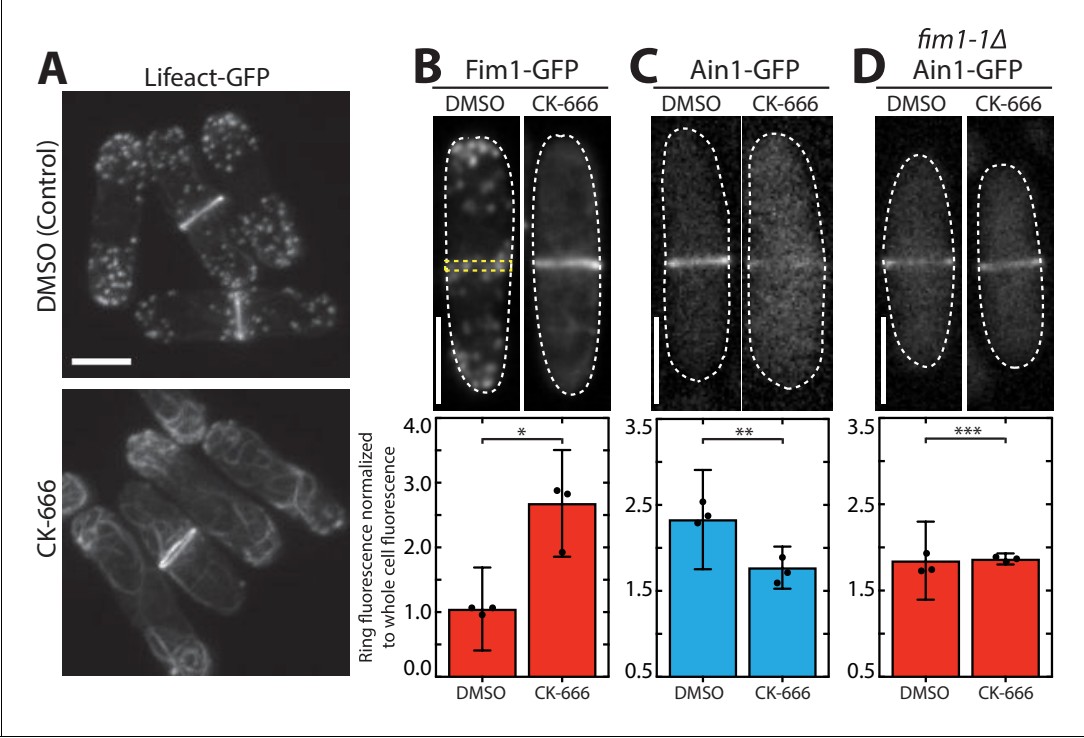

**Figure 1.** Fimbrin Fim1 and α-actinin Ain1 compete for association with the contractile ring. (A) Fluorescence micrographs of fission yeast cells expressing Lifeact-GFP following treatment with DMSO (control, top) or 200 µM Arp2/3 complex inhibitor CK-666 (bottom). Scale bar, 5 µm. (B-D, top panels) Fluorescence micrographs of fission yeast cells expressing Fim1-GFP (B), Ain1-GFP (C), or Ain1-GFP in a *fim1-1Δ* background (D), following treatment with DMSO (left) or 200 µM CK-666 (right). Dotted lines outline cells. Yellow dotted line denotes representative region used to quantify fluorescence value in cells lacking a visible contractile ring. Scale bars, 5 µm. (B-D, bottom panels) Mean Fim1-GFP (B) or Ain1-GFP (C,D) fluorescence at the contractile ring normalized to whole cell fluorescence. Error bars = s.d. Filled circles indicate means of experimental replicates. n ≥ 18 cells from three independent experiments. Two-tailed t-tests for data sets with unequal variance yielded p-values *p=8.57×10⁻²⁰, **p=1.75×10⁻⁶, ***p=0.81.

DOI: https://doi.org/10.7554/eLife.47279.002

The following figure supplements are available for figure 1:

**Figure supplement 1.** Contractile ring ABP localization following CK-666 treatment.
DOI: https://doi.org/10.7554/eLife.47279.003

**Figure supplement 2.** Tropomyosin Cdc8 does not leave the contractile ring following CK-666 treatment.
DOI: https://doi.org/10.7554/eLife.47279.004

**Figure supplement 3.** Fimbrin Fim1 displaces α-actinin Ain1 from the contractile ring following CK-666 treatment.
DOI: https://doi.org/10.7554/eLife.47279.005

patches in ~67% of *fim1-1Δ* cells (*Figure 2B,C*, *Figure 2—video 1*). Therefore, Fim1 and Ain1 appear to compete for association with F-actin at both actin patches and the contractile ring.

## Less dynamic α-actinin Ain1 associates with actin patches

Ultrastructural and mutational studies of fimbrin/plastin and α-actinin from several organisms demonstrate that they bind to a similar site on F-actin (*Galkin et al., 2010*; *Galkin et al., 2008*; *Holtzman et al., 1994*; *Honts et al., 1994*; *McGough et al., 1994*). However, while fission yeast fimbrin Fim1 is relatively stable on single actin filaments ($k_{off}$ = 0.043 ± 0.001 s⁻¹) and very stable on F-actin bundles ($k_{off}$ = 0.023 ± 0.003 s⁻¹) (*Skau et al., 2011*), α-actinin Ain1 has not been observed to associate with single actin filaments and is extremely dynamic on F-actin bundles ($k_{off}$ = 3.33 s⁻¹ on two-filament and three-filament bundles) (*Li et al., 2016*). Therefore, we hypothesized that Fim1's longer residence time on F-actin bundles may explain its ability to outcompete Ain1 for a similar F-actin binding site. To test this possibility, we took advantage of the Ain1 mutant Ain1(R216E), which is 5- to 10-fold less dynamic on F-actin bundles ($k_{off}$ = 0.67 s⁻¹ and $k_{off}$ = 0.33 s⁻¹ on two- and

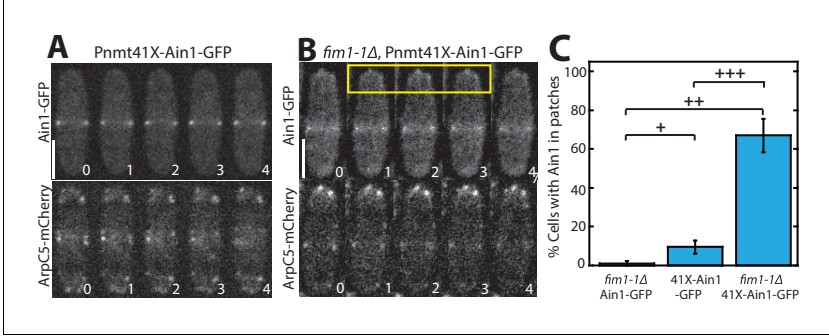

**Figure 2.** Fimbrin Fim1 and α-actinin Ain1 compete for association with actin patches. (A–B) Time-lapse fluorescent micrographs of fission yeast cells expressing ArpC5-mCherry (bottom) and overexpressing GFP-tagged α-actinin Ain1 from the 41Xnmt promoter (top) for 20 hr in a wild-type (A) or *fim1-1Δ* background (B). Yellow box highlights Ain1-GFP localization at actin patches. Scale bars, 5 μm. Time in s. (C) Percentage of cells in which Ain1-GFP is observed in actin patches. Error bars = s.e. Two-tailed t-tests for data sets with unequal variance yielded p-values+p = 0.113,+$^+$p = 0.002,+$^{++}$p = 0.012. n = 3 experimental replicates.

DOI: https://doi.org/10.7554/eLife.47279.006

The following video and figure supplement are available for figure 2:

**Figure supplement 1.** Cellular expression of GFP-tagged α-actinin Ain1 constructs.

DOI: https://doi.org/10.7554/eLife.47279.007

**Figure 2—video 1.** Overexpressed α-actinin Ain1-GFP localizes to actin patches in the absence of fimbrin, related to *Figure 1*.

DOI: https://doi.org/10.7554/eLife.47279.008

three-filament bundles, respectively) (*Li et al., 2016*), and assessed its ability to compete with Fim1 in vitro and in vivo.

We utilized multi-color TIRF microscopy (TIRFM) to directly visualize the association of fluorescently labeled Fim1 with actin filaments in vitro in either the presence or absence of unlabeled Ain1. Compared to 50 nM Fim1-TMR alone, which fully decorates F-actin bundles, less Fim1-TMR associates with two filament F-actin bundles in the presence of either 1 μM wild-type Ain1 or Ain1(R216E) (*Figure 3A,B*). However, there is little difference in the amount of Fim1-TMR associated with two-filament F-actin bundles in the presence of wild-type Ain1 or mutant Ain1(R216E).

Although we did not detect that Ain1(R216E) competes with Fim1 better than does wild-type Ain1 in vitro (*Figure 3*), the less dynamic Ain1(R216E) mutant is better than wild-type Ain1 at associating with Fim1-bound actin patches in vivo (*Figure 4*). In fission yeast cells expressing endogenously tagged Fim1-mCherry, overexpressed wild-type Ain1-GFP localizes to actin patches in only ~9% of cells (*Figure 4A,C*, *Figure 4—video 1*), whereas overexpressed mutant Ain1(R216E)-GFP localizes to actin patches in 100% of cells (*Figure 4B,C*, *Figure 4—video 1*). The disparity between Ain1(R216E)'s ability to compete with Fim1 in vitro versus in vivo potentially suggests that slight differences in dynamics may have a bigger effect in a cellular context. In particular, the dynamics of ABPs such as Ain1 may be finely tuned to allow for proper sorting given the large number of actin interacting proteins, with a small change in dynamics skewing the sorting to a dramatic degree. Alternatively, it is possible that our simplified in vitro system does not fully mimic in vivo conditions.

## Tropomyosin Cdc8 and α-actinin Ain1 do not compete for association with actin filaments

We previously reported that tropomyosin Cdc8, an F-actin side-binding protein that associates with the contractile ring, is displaced from F-actin by fimbrin Fim1 in vitro and is thereby prevented from associating with actin patches in fission yeast cells (*Christensen et al., 2017*; *Skau and Kovar, 2010*). Although fimbrin/plastin and α-actinin isoforms bind to a similar site on F-actin, Cdc8 and α-actinin Ain1 both associate with the contractile ring. Therefore, we considered whether Ain1 also displaces Cdc8 from F-actin, or if they can simultaneously associate with F-actin. In multi-color in vitro TIRFM assays, Cdc8 is not displaced from Ain1- or Ain1(R216E)-bundled F-actin networks (*Figure 5A,B*, *Figure 5—video 1*), but it is displaced from Fim1-bundled networks (*Figure 5C*;

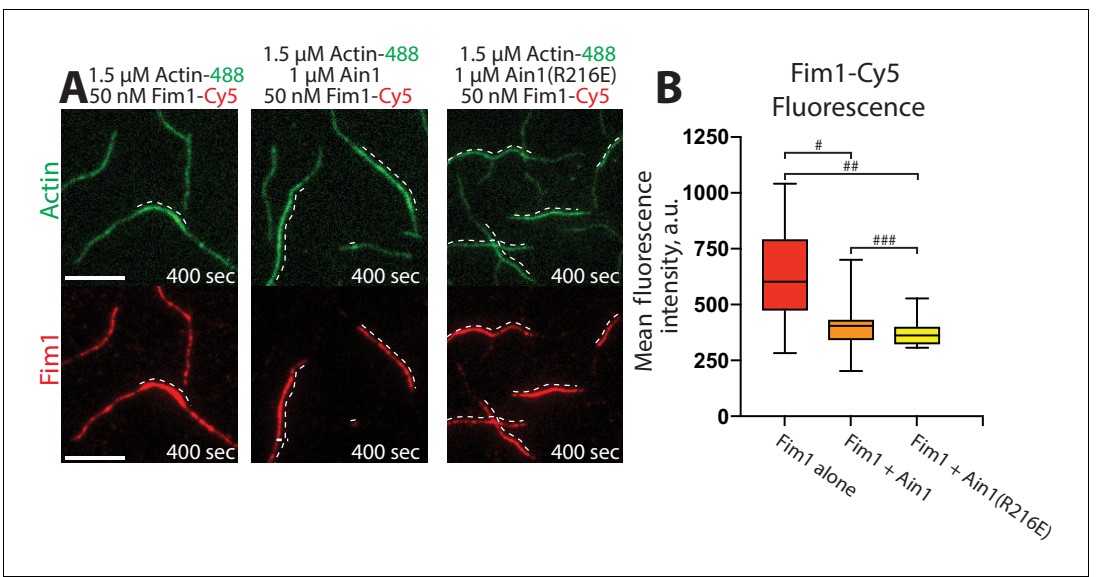

**Figure 3.** Fimbrin Fim1 and α-actinin Ain1 compete for F-actin binding F-actin in vitro. (A–B) Two-color TIRFM of 1.5 μM Mg-ATP actin (15% Alexa 488-labeled) with 50 nM fimbrin Fim1 (Cy5-labeled) alone, or with either 1 μM wild-type α-actinin Ain1 or mutant Ain1(R216E). Scale bars, 5 μm. Dotted lines denote bundled regions. (B) Box plots of the amount of Fim1-Cy5 fluorescence on two-filament bundles in either the absence (red) or presence of Ain1 (orange) or Ain1(R216E) (yellow). Error bars = s.e. Two-tailed t-tests for data sets with unequal variance yielded p-values #p=3.90×10$^{-4}$, ##p=1.97×10$^{-5}$, and ###p=0.18. Two independent experiments were performed for each condition. In total, n ≥ 16 two-filament bundle measurements were taken for each condition.
DOI: https://doi.org/10.7554/eLife.47279.009

*Christensen et al., 2017*; *Skau and Kovar, 2010*). Thus, Ain1 and Cdc8 are capable of co-existing on the same F-actin network in vitro as they do at the contractile ring in cells.

## Tropomyosin Cdc8 enhances the bundling activity of α-actinin Ain1

Compared to fimbrin Fim1, α-actinin Ain1 is a relatively weak F-actin bundling protein (*Addario et al., 2016*; *Li et al., 2016*; *Morita et al., 2017*). Remarkably, low-speed sedimentation (*Figure 6A,B*) and TIRFM assays (*Figure 6C,D*, *Figure 6—video 1*) revealed that tropomyosin Cdc8 significantly enhances the F-actin bundling ability of Ain1. In single-color TIRFM assays with unlabeled ABPs, 500 nM Cdc8 increases Ain1-mediated F-actin bundling 10-fold over Ain1 alone (*Figure 6D*, *Figure 6—video 1*). This surprising finding suggests that the combination of Ain1 and Cdc8 may allow for significant F-actin bundling to occur in the context of the contractile ring despite Ain1's poor bundling ability alone.

Cdc8's enhancement of Ain1-mediated F-actin bundling could potentially arise from Cdc8 increasing the number of Ain1 binding events on F-actin. To test this possibility, we analyzed the single molecule dynamics of Ain1 by performing TIRFM experiments with sparsely-labeled (0.5% TMR-labeled) Ain1 on uncoated versus Cdc8-coated F-actin. Three-fold more Ain1 binding events were observed on Cdc8-coated F-actin compared to uncoated F-actin (*Figure 6E–G*), suggesting that Cdc8 enhances the binding of Ain1 to F-actin, thereby increasing the F-actin bundling ability of Ain1.

## Tropomyosin Cdc8 and α-actinin Ain1 cooperate to displace fimbrin Fim1 from actin filaments

On their own, both α-actinin Ain1 and tropomyosin Cdc8 are outcompeted by fimbrin Fim1 for binding to F-actin. Furthermore, there are ~86,500 Fim1 polypeptides in the cell, but only ~3600 Ain1 molecules (*Wu and Pollard, 2005*), raising the question as to why Fim1 is not associated more strongly with the contractile ring in wild-type cells. Given that Cdc8 enhances the bundling ability of Ain1, we speculated that the combination of Cdc8 and Ain1 might inhibit Fim1 association with actin

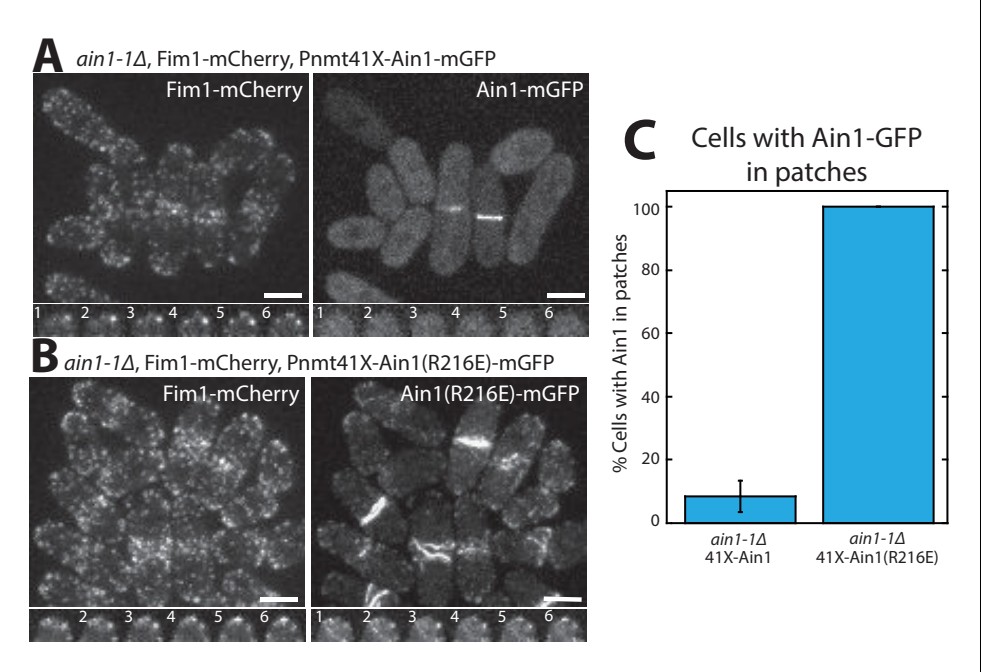

**Figure 4.** Fimbrin Fim1 and α-actinin Ain1 competition at actin patches is driven by their residence time on F-actin. (A,B, top) Fluorescence micrographs of fission yeast in an *ain1-1Δ* background overexpressing GFP-tagged wild-type Ain1 (A) or mutant Ain1(R216E) (B) from the 41Xnmt1 promoter. Scale bars, 5 μm. (A,B, bottom) Timelapse (in s) of cell taken from a single Z-plane. (C) Percentage of cells in which Ain1-GFP is observed in actin patches. Error bar, s.e. Two-tailed t-test for data sets with unequal variance yielded p-value=0.0029.
DOI: https://doi.org/10.7554/eLife.47279.010
The following video is available for figure 4:
**Figure 4—video 1.** Overexpressed mutant α-actinin Ain1(R216E)-GFP, but not Ain1-GFP, localizes to actin patches in the presence of fimbrin Fim1, related to *Figure 4*.
DOI: https://doi.org/10.7554/eLife.47279.011

filaments. To test this possibility, we first performed low speed F-actin sedimentation assays in the presence of Cdc8 and either Fim1, Ain1, or both (*Figure 7A,B*). Almost two-fold more Cdc8 pellets with Ain1-mediated F-actin bundles than with Fim1-mediated bundles (*Figure 7B*). Additionally, in the presence of both Fim1 and Ain1, an intermediate amount of Cdc8 pellets in the presence of both Fim1 and Ain1, suggesting that Ain1 allows Cdc8 to better associate with F-actin in the presence of Fim1.

To directly investigate the effect of Ain1 and Cdc8 cooperation on competition with Fim1, we performed four-color TIRFM with fluorescently labeled ABPs and quantified Fim1 association with F-actin in the presence of Cdc8 and/or Ain1. In the absence of Ain1, Cdc8 is displaced from F-actin bundles by Fim1 in a cooperative manner, with most F-actin bundles completely devoid of Cdc8, concurrent with regions of high Fim1 localization (*Figure 8A*, *Figure 8—video 1*; *Christensen et al., 2017*). Conversely, in reactions containing Ain1, over 100-fold more Cdc8 is associated with F-actin bundles (*Figure 8B,C*, *Figure 8—video 1*). Furthermore,~40% less Fim1 is observed to associate with these bundles (*Figure 8B,D*, *Figure 8—video 1*), suggesting that the combination of Ain1 and Cdc8 is capable of inhibiting Fim1 association with F-actin networks. Additionally, in the presence of Cdc8, Fim1 and Ain1 sort into mutually exclusive domains along F-actin bundles (*Figure 8B*), similar to what has been observed previously between human Fascin1 and α-actinin-4 (*Winkelman et al., 2016*). We speculate that competition for the same binding site allows Ain1 to prevent long stretches of Fim1 from forming that might be capable of displacing Cdc8. Thus, Fim1 poorly associates with F-actin in the presence of both Cdc8 and Ain1 because 1) Cdc8 and Ain1 prevent Fim1 from cooperatively associating with F-actin and 2) Cdc8 and Ain1 synergize to compete with Fim1.

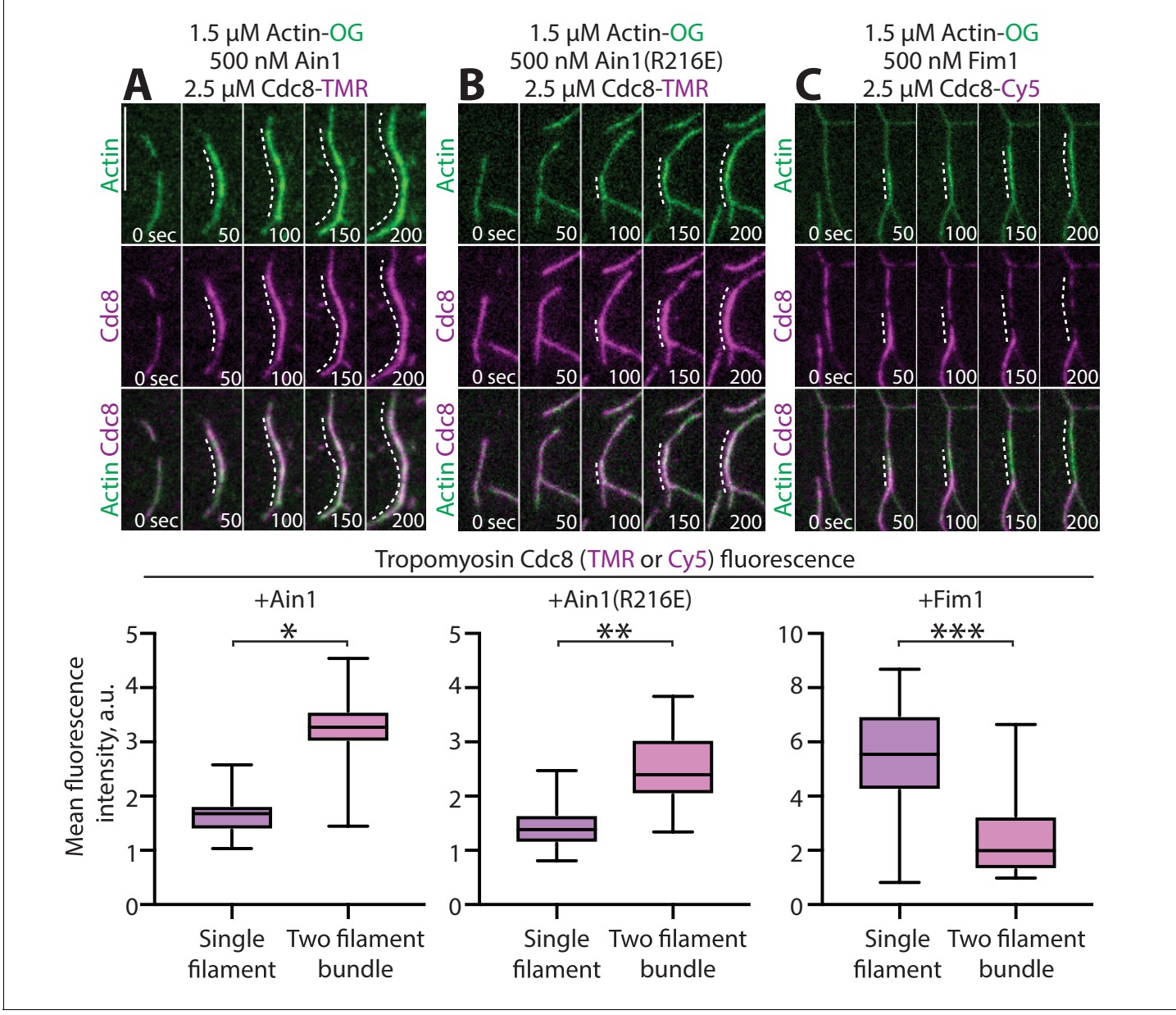

**Figure 5.** α-actinin Ain1 does not displace tropomyosin Cdc8 from F-actin bundles in vitro. (A-C, top) Two-color TIRFM of 1.5 μM Mg-ATP actin (15% Alexa 488-labeled) with 2.5 μM tropomyosin Cdc8 (TMR-labeled) and unlabeled 500 nM (**A**) wild-type α-actinin Ain1, (**B**) mutant Ain1(R216E), or (**C**) fimbrin Fim1. Scale bar, 1 μm. Dotted lines denote bundled regions. (A-C, bottom) Dot plots of the amount of Cdc8-TMR or Cdc8-Cy5 fluorescence on single filaments or two-filament bundles in the presence of Ain1 (**A**), Ain1(R216E) (**B**) or Fim1 (**C**). Error bars = s.e. Two-tailed t-tests for data sets with unequal variance yielded p-values *p=8.24×10$^{-18}$, **p=5.47×10$^{-12}$, ***p=5.72×10$^{-11}$.

DOI: https://doi.org/10.7554/eLife.47279.012

The following video is available for figure 5:

**Figure 5—video 1.** α-actinin Ain1 does not displace tropomyosin Cdc8 from F-actin bundles, related to *Figure 5*.

DOI: https://doi.org/10.7554/eLife.47279.013

The synergy between Cdc8 and Ain1 may explain why Fim1 only poorly associates with contractile rings in fission yeast cells (*Figure 8E*).

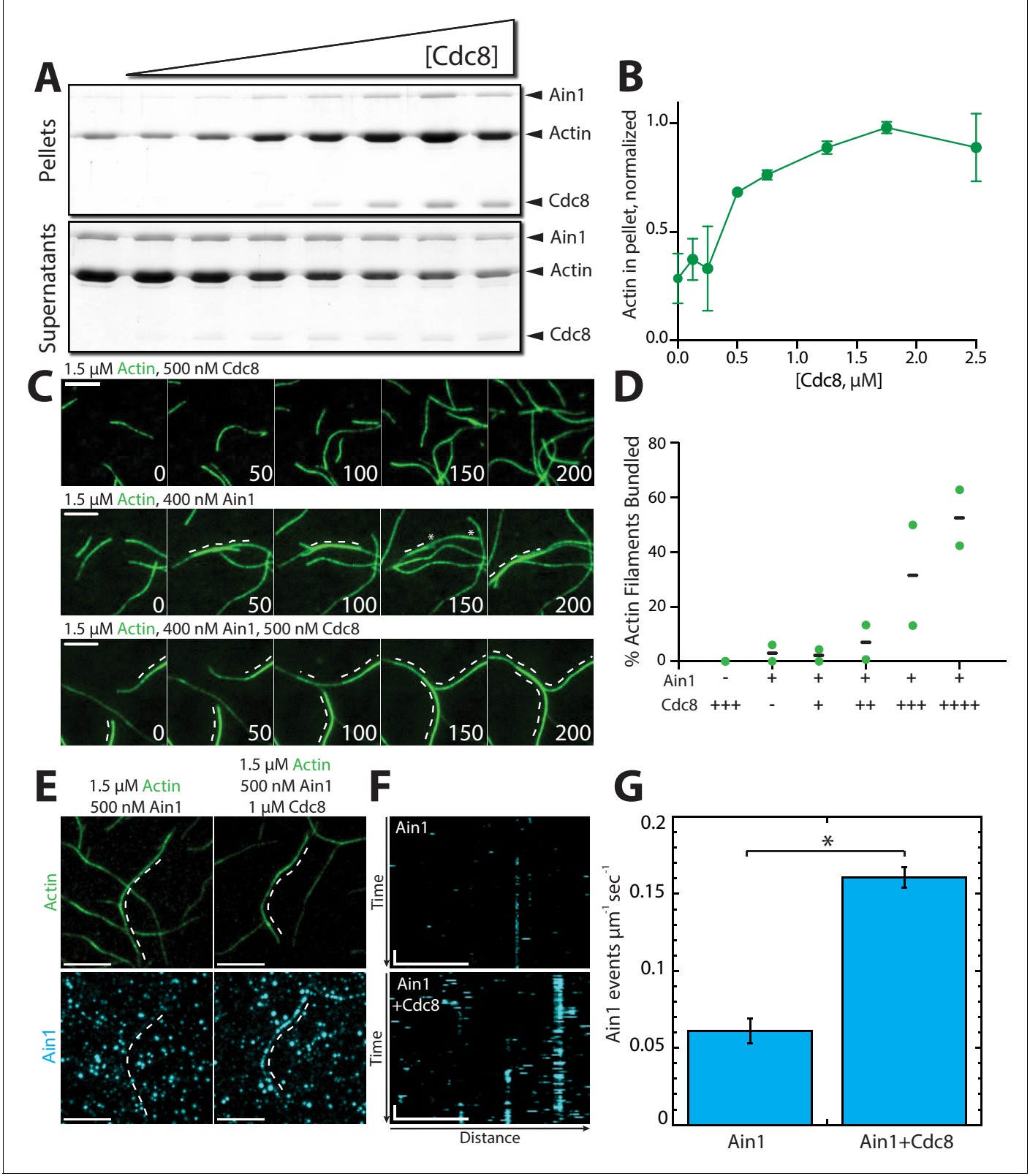

**Figure 6.** Tropomyosin Cdc8 enhances α-actinin Ain1-mediated F-actin bundling in vitro. (**A–B**) Low-speed (10,000 x *g*) sedimentation assays of 5 μM Mg-ATP preassembled actin filaments, 500 nM Ain1, and increasing concentrations of Cdc8 (0–5 μM). (A) Coomassie blue-stained gel of pellets and supernatants from a representative experiment. (B) Quantification of actin extracted from the pellet as a function of Cdc8 concentration. Error bars = s. d, n = 2. (**C-D**) TIRFM of 1.5 μM Mg-ATP actin (15% Alexa 488-labeled) in the presence of 500 nM Cdc8 (C, top), 400 nM Ain1 (C, middle) or both (C,

*Figure 6 continued on next page*

*Figure 6 continued*

bottom). Dotted lines indicate the bundled region. Scale bars, 5 µm. (D) Quantification of the percent of bundled F-actin with 500 nM Cdc8 alone, 400 nM Ain1 alone, or 400 nM Ain1 with 50 nM, 125 nM, 250 nM or 500 nM Cdc8. Black lines indicate averages and green circles indicate values from independent TIRFM experiments. n = 2 independent experiments for each condition. (E-G) Spot density TIRFM experiments of 1.5 µM Mg-ATP actin (15% Alexa 488-labeled) and 500 nM Ain1 (0.5% TMR-labeled) alone or with 1 µM unlabeled Cdc8. (E, top) Representative images of Alexa-488-labeled F-actin bundles. (E, bottom) Representative max projection of all Ain1 spots on corresponding F-actin bundles. Scale bars, 5 µm. (F) Kymographs of the indicated bundle from (E) over time. Scale bar, 5 µm. Time bar, 11 s. (G) Ain1 spot density (Ain1 events/µm/s). Two-tailed t-tests for data sets with unequal variance yielded p-value *p=0.026. Error bars = s.e. n = 2 independent experiments.

DOI: https://doi.org/10.7554/eLife.47279.014
The following video is available for figure 6:
**Figure 6—video 1.** Tropomyosin Cdc8 enhances α-actinin Ain1-mediated bundling, related to *Figure 6*.
DOI: https://doi.org/10.7554/eLife.47279.015

## Discussion

We have discovered that fimbrin Fim1 is inhibited from associating with actin filaments by the combined efforts of contractile ring ABPs α-actinin Ain1 and tropomyosin Cdc8. Although we have demonstrated that competition between ABPs is one key mechanism that drives their sorting to different F-actin networks, we suspect that there are several non-mutually exclusive mechanisms that drive the sorting of Fim1 and other ABPs to the correct F-actin network. Specifically, our data suggest that Fim1's association with the contractile ring is inhibited in part by both (1) a preferred association with actin patches and (2) competition with Cdc8 and Ain1.

### A 'sink' model for fimbrin Fim1 association with actin patches

If fimbrin Fim1 preferentially associates with actin patches over other F-actin networks, actin patches could act as a 'sink' for Fim1, thereby sequestering Fim1 and limiting the concentration of free Fim1 capable of associating with other F-actin networks. Indeed, although only small amounts of Fim1 are present on the contractile ring of wild-type cells (*Nakano et al., 2001*; *Wu et al., 2001*), depletion

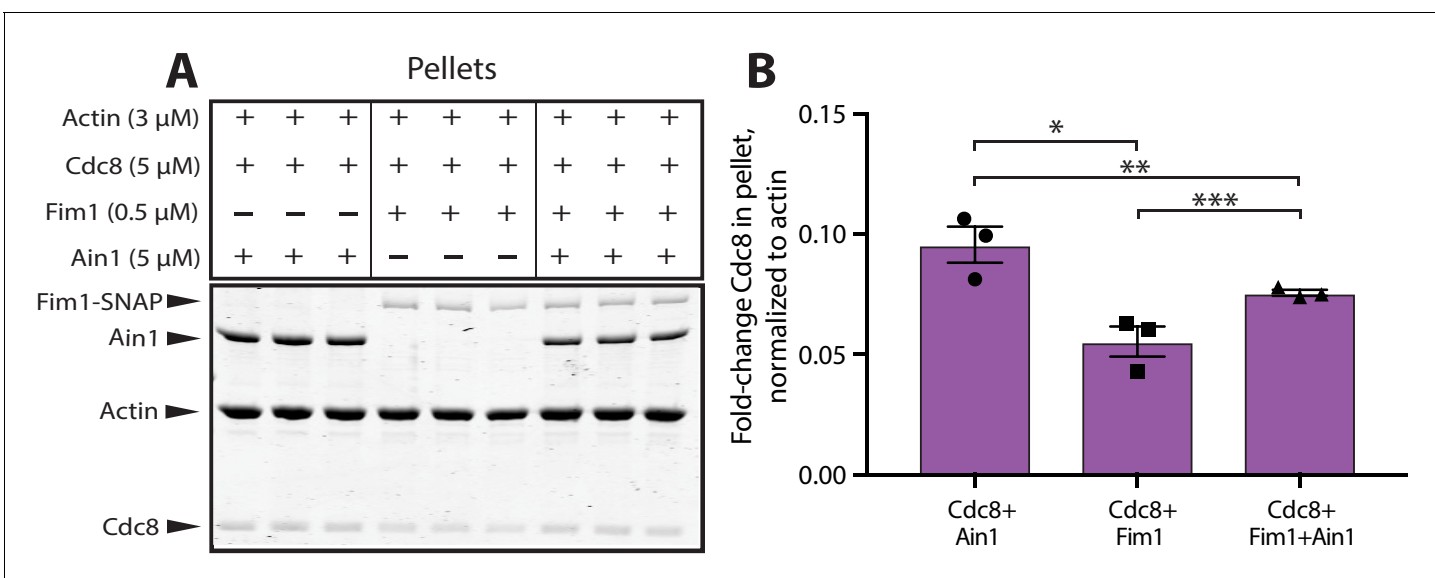

**Figure 7.** α-actinin Ain1 facilitates the association of tropomyosin Cdc8 with bundled F-actin in the presence of fimbrin Fim1 in vitro. (A,B) Low-speed sedimentation comparing Cdc8 in the pellet with F-actin bundled by Fim1 (left), Ain1 (middle), or both (right). (A) Coomassie blue-stained gel of representative triplicate samples of 3 µM Mg-ATP pre-assembled actin filaments incubated and centrifuged with fixed concentrations of Cdc8 (5 µM), Fim1 (0.5 µM), and Ain1 (5 µM). (B) Fold change of Cdc8 in the pellet from conditions presented in (A). Error bars = s.e, n = 3. Filled shapes indicate individual values from each of three replicates as seen in (A). Two-tailed t-tests for data sets with unequal variance yielded p-values *p=0.0450, **p=0.629, ***p=0.0348.

DOI: https://doi.org/10.7554/eLife.47279.016

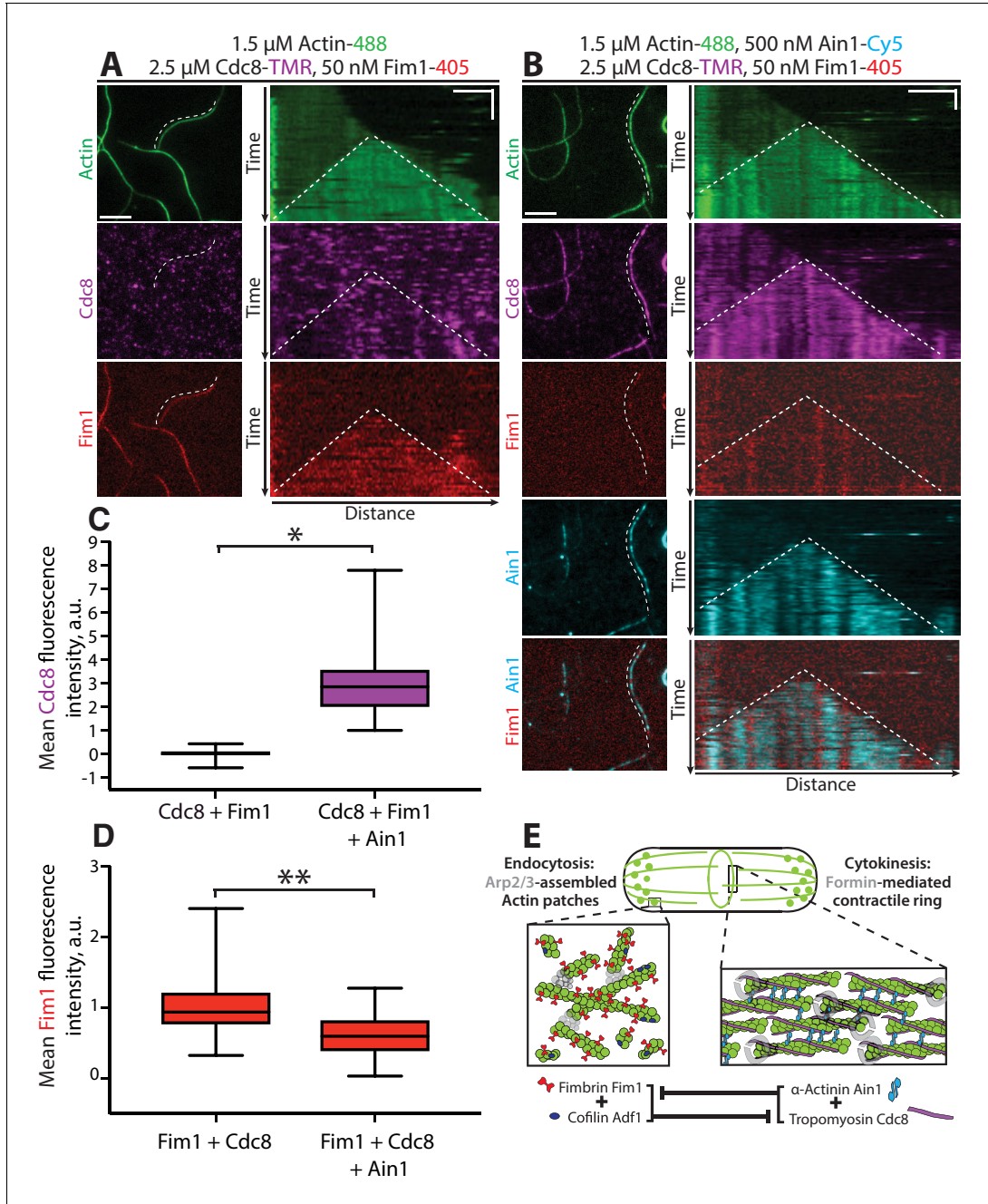

**Figure 8.** Tropomyosin Cdc8 and α-actinin Ain1 cooperate to compete with fimbrin Fim1 for association with F-actin in vitro. (A–D) Four-color TIRFM of 1.5 μM Mg-ATP actin (15% Alexa 488-labeled) with 50 nM fimbrin Fim1 (Alexa 405-labeled) and 2.5 μM tropomyosin Cdc8 (TMR-labeled) in the (A) absence or (B) presence of 500 nM α-actinin Ain1 (Cy5-labeled). (A-B, left) Representative TIRF field. Dotted lines denote bundled regions. Scale bar, 5 μm. (A-B, right) Kymographs of actin, Fim1, and Cdc8 during bundle formation. Dotted lines denote bundled regions. Scale bars, 3 μm. Time bar, 1 min. (C–D) Box plots of the amount of Cdc8-TMR (C) or Fim1-Cy5 fluorescence (D) on two-filament bundles in experiments with Cdc8 and Fim1, or Cdc8, Fim1 and Ain1. Two-tailed t-tests for data sets with unequal variance yielded p-values *p=$3.88\times10^{-16}$ and **p=$7.09\times10^{-7}$. n ≥ 30 measurements from two independent experiments. (E) Model of the involvement of ABP competition in ABP sorting in the fission yeast cell. In endocytic actin patches, fimbrin Fim1 and cofilin Adf1 enhance each other's activities, resulting in the displacement of tropomyosin Cdc8 from the F-actin network (*Christensen et al., 2017*). In the contractile ring, α-actinin Ain1 and tropomyosin Cdc8 work together to prevent fimbrin Fim1 association with the F-actin network.

DOI: https://doi.org/10.7554/eLife.47279.017

The following video is available for figure 8:

**Figure 8—video 1.** Tropomyosin Cdc8 and α-actinin Ain1 cooperate to compete with fimbrin Fim1 for bundling F-actin, related to *Figure 5*.

*Figure 8 continued on next page*

Figure 8 continued

DOI: https://doi.org/10.7554/eLife.47279.019

of actin patches (the 'sink') by the Arp2/3 complex inhibitor CK-666 results in significantly more Fim1 associating with the contractile ring (*Figure 1B*). However, a sink model alone likely cannot account for Fim1's primary localization to actin patches. Only 30% (~25,000) of the ~86,500 total Fim1 molecules in a fission yeast cell associate with ~50 actin patches (*Wu and Pollard, 2005*; *Sirotkin et al., 2010*). Therefore, ~70–75% of Fim1 molecules are either cytoplasmic or associated with another F-actin network. We suspect that other mechanisms (conformational changes in the actin filament, ABP dynamics, post-translational modifications) may dictate both Fim1's preferential association with actin patches, as well as the observed competitive and cooperative interactions amongst Fim1, α-actinin Ain1, and tropomyosin Cdc8.

## Conformational changes in the actin filament

Actin forms a flexible polymer capable of adopting a variety of conformations (*Galkin et al., 2010*; *Oda and Maéda, 2010*), and different F-actin conformations have been demonstrated to both result from or influence the association of ABPs (*Michelot and Drubin, 2011*; *Papp et al., 2006*; *Risca et al., 2012*). Fimbrin Fim1's inherent preference for actin patches could result from a preference for either a branched F-actin network or for a particular twist or conformational change imparted by Arp2/3 complex-mediated actin assembly. We suspect that conformational changes in the actin filament may also affect the observed increase of α-actinin Ain1-mediated bundling in the presence of tropomyosin Cdc8 (*Figure 6A–D*). Tropomyosins have been found to increase the persistence length of F-actin (*Fujime and Ishiwata, 1971*; *Isambert et al., 1995*), suggesting that they are capable of altering the conformation of an actin filament. Therefore, Cdc8 may mediate a slight conformational change more amenable to Ain1 binding, increasing the density of Ain1 on the actin filaments and the corresponding degree of bundling. Tropomyosin's increase of the persistence length of F-actin may have an additional effect on F-actin bundling by promoting bundle stability and inter-filament contacts.

## ABP dynamics affect their ability to compete

We discovered that the contractile ring ABP α-actinin Ain1 competes with fimbrin Fim1, and that dynamic association of both Ain1 and Fim1 with actin filaments affects their ability to compete at different F-actin networks. In particular, we found that a less dynamic Ain1(R216E) mislocalizes to F-actin patches even in the presence of Fim1. However, in addition to ectopic localization to actin patches, cells expressing Ain1(R216E) also have cytokinesis defects (*Li et al., 2016*). These findings may explain why fission yeast requires two F-actin bundling proteins. Actin patches require a stable F-actin bundler such as Fim1 to crosslink the branched F-actin network, prevent tropomyosin Cdc8 association, and create boundaries that enhance cofilin Adf1-mediated severing (*Christensen et al., 2017*; *Skau and Kovar, 2010*). On the other hand, a dynamic F-actin bundler is required at the contractile ring to facilitate anti-parallel F-actin contacts while still allowing Cdc8 association and myosin-mediated actin filament sliding and contraction (*Li et al., 2016*). For these reasons, we assume that the dynamic Ain1 is an ideal actin crosslinker for the contractile ring (*Figure 6E–G*). Though our results suggest that Ain1 is more dynamic on uncoated actin filaments, we showed that wild-type Ain1 is still a dynamic bundler on Cdc8-coated actin filaments and is likely capable of allowing contractile ring ABPs to remain associated with and function optimally at the contractile ring.

Additionally, the binding dynamics of Ain1 and Fim1 on F-actin seem to mediate their competition with Cdc8. We demonstrated that Fim1, but not Ain1, displaces Cdc8 from F-actin bundles. However, Fim1 competes with Cdc8 specifically at regions where it binds stably (F-actin bundles), and does not compete as strongly with Cdc8 on single filaments (*Christensen et al., 2017*). Therefore, the presence of a dynamic bundling protein (Fim1 on single filaments and Ain1 on both single and bundled filaments) may allow Cdc8 to remain associated with F-actin in those circumstances. Future studies will seek to build a fuller picture of how Fim1 and Ain1 differentially affect the activity of other ABPs present at the contractile ring such as myosin-II and Adf1.

## Post-translational modifications and other sorting mechanisms

In vitro, the combined efforts of α-actinin Ain1 and tropomyosin Cdc8 prevent ~40% of fimbrin Fim1 association with F-actin bundles. It should be noted that although Ain1 and Cdc8 work together to compete with Fim1, our reactions contain low concentrations of Fim1 (50 nM) compared to Ain1 (500 nM or 1 μM) and Cdc8 (2.5 μM). Given the potent F-actin binding and bundling abilities of Fim1 and the high cellular concentration of Fim1, a reasonable assumption is that Cdc8 and Ain1 may only prevent a portion of Fim1 polypeptides from associating with contractile ring F-actin in vivo. Therefore, other non-mutually exclusive mechanisms likely contribute to ABP sorting. One possibility is that additional ABPs inhibit Fim1 from associating with the contractile ring. A second possibility is that Fim1 is post-translationally modified, regulating its activity. Budding yeast fimbrin Sac6 is phosphorylated at different stages of the cell cycle, which affects its ability to bind F-actin (*Miao et al., 2016*). Fission yeast Fim1 might be similarly post-translationally modified (*Kettenbach et al., 2015*; *Swaffer et al., 2018*), and therefore a portion of the cytoplasmic Fim1 pool might be more or less active. A third possibility is that mechanical stresses applied to actin filaments, such as the contractile forces applied by the molecular motor type-II myosin Myo2 during contractile ring assembly and constriction, might differentially alter the F-actin binding and/or activity of actin patch and contractile ring ABPs (*Romet-Lemonne and Jégou, 2013*).

## A similar combination of mechanisms could drive ABP sorting to other F-actin networks

We expect that similar fundamental mechanistic principles also promote ABP sorting at other F-actin networks, both in fission yeast and in multicellular organisms. One possibility is that actin assembly factors, such as the Arp2/3 complex (actin patches), formin Cdc12 (contractile rings), and formin For3 (actin cables) assemble actin filaments with intrinsic differences (such as a specific filament conformation) that initiate the differential sorting of certain ABPs to diverse networks (*Kovar et al., 2011*; *Michelot and Drubin, 2011*). The initial actin assembly factor-dependent recruitment of 'upstream' ABPs may then subsequently recruit and/or inhibit other ABPs, defining both the subset of ABPs that associate with a particular F-actin network as well as the network's corresponding architecture and dynamics. For example, although fission yeast has a single tropomyosin isoform, Cdc8, it can be either acetylated or unacetylated (*Skoumpla et al., 2007*). While the acetylated form of Cdc8 associates with the contractile ring, the unacetylated form associates with actin cables (*Coulton et al., 2010*). Unacetylated Cdc8 preferentially associates with For3-assembled F-actin regardless of where it is assembled (*Johnson et al., 2014*). We used a contractile ring acetylation mimic form of Cdc8 (AlaSer-Cdc8) in the in vitro assays reported here (*Christensen et al., 2017*). However, it is possible that the acetylated and unacetylated forms of Cdc8 differentially compete or cooperate with other ABPs in vitro and/or in vivo, influencing the sorting of distinct sets of ABPs to each network. Interestingly, how ABPs influence one another could also depend on the cellular type or context, as tropomyosin and α-actinin-4 exhibit an antagonistic relationship during stress fiber formation in MDCK cells (*Kemp and Brieher, 2018*). Future work will involve investigating the potential role of actin assembly factors in regulating ABP sorting and understanding the competitive and cooperation interactions between ABPs in other cell types.

## Materials and methods

### Strain construction and growth

Fission yeast strains were created by genetic crossing on SPA5S plates followed by tetrad dissection on YE5S plates. Strains were screened for auxotrophic (leu, ura) or antibiotic (nat, kan) markers and maintained on YE5S plates. Glycerol stocks were created by pelleting cells and resuspending in 750 μL media and 250 μL of 50% sterile glycerol. The strains used in this study are indicated in *Table 1*.

### Cell imaging and treatment with CK-666

For live cell imaging, cells were grown in YE5S media overnight at 25°C, subcultured into EMM5S media without thiamine, and kept in log phase for 20–22 hr at 25°C. Cells were imaged directly on glass slides. Z-stacks of 10 slices, 0.5 μm apart were acquired with a 100x, 1.4 NA objective on a Zeiss Axiovert 200M equipped with a Yokogawa CSU-10 spinning-disk unit (McBain, Simi Valley, CA)

**Table 1.** Fission yeast strains used in this study.

| Strain name | Genotype | Reference |
| --- | --- | --- |
| KV91 | h+, kanMX6::myo2p::gfp-myo2p+, ade6-M210, leu1-32, ura4-D18 | *Wu et al., 2003* |
| KV343 | h?, cdc12-mGFP::KanR | This study |
| KV459 | h+, rlc1-tdTomato-natMX6 ade6-M210 leu1-32 ura4-D18 | This study |
| KV579 | h-, ain1-GFP-KanMX6, URA+ | *Wu et al., 2001* |
| KV588 | h+, pAct1 Lifeact-GFP::Leu+; ade6-m216; leu1-32; ura4-D18 | *Huang et al., 2012* |
| *Laporte et al., 2011* KV707 | h-, leu1-32, his3-D1, ura4-D18, ade6-M216, Pnmt41-SpAin1-mGFP::Leu+ | This study |
| KV804 | h?, fim1-mCherry-natMX6, ain1-Δ1:: kanMX6, Pnmt41-SpAin1-mGFP::Leu+ | This study |
| KV818 | h + kanMX6-Prng2-mEGFP-rng2 ade6-M210 leu1-32 ura4-D18 | *Laporte et al., 2011* |
| KV856 | h?, ain1-Δ1:: kanMX6, Pnmt41-SpAin1(R216E)-mGFP::Leu+, fim1-mCherry-natMX6, ade6, leu1-32, ura4-D18 | This study |
| KV861 | h?, ain1-GFP-kanMX6, sad1-tdTomato-natMX6, ade6-m21?, leu1-32, ura4-D18 | This study |
| KV878 | h+, fim1-mGFP-kanMX6, sad1-tdTomato-natMX6 | This study |
| KV908 | h? fim1-1Δ-kanMX6, ain1-GFP-kanMX6, sad1-tdTomato::natMX6 | This study |
| KV963 | h?, fim1-1Δ::kanMX6, Pnmt41-SpAin1-mGFP::Leu+ | This study |
| KV968 | h? Pnmt41-SpAin1-mGFP::Leu+, ARPC5-mCherry-natMX6 | This study |

DOI: https://doi.org/10.7554/eLife.47279.018

illuminated with a 50-milliwatt 473 nm DPSS laser, and a Cascade 512B EM-CCD camera (Photometrics, Tucson, AZ) controlled by MetaMorph software (Molecular Devices, Sunnyvale, CA). For CK-666 treatments, CK-666 powder stock (Sigma, St. Louis, MO) was diluted to 10 mM in DMSO. Cells were grown as stated above, and incubated with CK-666 or an equivalent volume of DMSO (control) in a rotator at 25°C for 30 min prior to imaging. Cells were then immediately imaged as above.

## Contractile ring fluorescence quantification

Contractile ring maturation was divided into three stages by measuring the distance between spindle pole bodies (SPBs, visualized by Sad1-tdTomato) and noting constriction of the contractile ring. Stage 1 cells had SPBs less than 6 μm apart, with no observable ring constriction. Stage 2 cells had SPBs greater than 6 μm apart, with no observable ring constriction. Stage 3 cells had SPBs less than 9 μm apart, with evident ring constriction. Quantification of ABP association with all contractile rings (stages 1–3) is shown in *Figure 1B–D*, and quantification at each distinct ring stage is show in *Figure 1—figure supplement 3*. The contractile ring region was determined by visually examining the z-stack for the ring site. Normalized contractile ring fluorescence was taken by drawing a region of interest (ROI) around the observed ring and around the whole cell using ImageJ. To obtain a fluorescence value for cells without a visible contractile ring (for example, DMSO-treated Fim1-GFP cells), mitotic cells were determined by the presence of two spindle pole body markers. These fission yeast cells were measured along their length and a 4-pixel width line was drawn across the exact center of the fission yeast cell (denoted as a yellow dotted-line in *Figure 1B*). The mean fluorescence of the ring divided by the whole cell was then determined. A value of 1.00 indicates no increased fluorescence at the site of the contractile ring, while values > 1 indicate increased fluorescence at the ring. Maximum projections were used for images in figures and sum projections were used for quantification.

## Tropomyosin Cdc8 antibody staining

Following standard growth and culturing protocols for live cell imaging, fission yeast cells were stained with anti-Cdc8p (*Cranz-Mileva et al., 2015*). Cells were fixed in 16% formaldehyde for 5 min at 20°C. Cells were then washed in cold 1X PBS and resuspended in 140 μL 1.2M sorbitol. 60 μL fresh protoplasting solution (3 mg/ml zymolase 100T in 1.2M sorbitol) was added and cells were incubated for 7 min on a rotator at room temperature. 1 mL of 1% Triton-X was then added to the

cells and incubation continued for 2 min. Cells were then pelleted and resuspended in 0.5 mL PBAL (10% BSA, 100 mM lysine monohydrochloride, 1 mM $NaN_3$, 50 ng/ml ampicillin in PBS) and incubated for 2.5 hr on a rotator at room temperature. Cells were resuspended in 100 µL of anti-Cdc8p 1:10 in PBAL (gift of Sarah Hitchcock-DeGregori) and incubated overnight at 4°C on a rotator. Following incubation with primary antibody, cells were washed three times with 0.5 mL PBAL and resuspended in 50 µL Alexa-Flour 555 goat anti-rabbit secondary antibody (Thermo-Fisher Scientific, Carlsbad, CA) (1:100 in PBAL) and incubated for 90 min at room temperature on a shaker in the dark. Cells were then washed five times with 0.5 mL PBAL and resuspended in 20–30 µL PBAL for imaging. Cells were stored at 4°C and imaged within 48 hr of staining.

## Protein purification

Chicken skeletal muscle actin was purified as described previously (*Spudich and Watt, 1971*). Fimbrin Fim1 and acetylation mimic tropomyosin AlaSer-Cdc8 (WT and I76C mutant) were expressed in BL21-Codon Plus (DE3)-RP (Agilent Technologies, Santa Clara, CA). His-tagged Fim1 was purified using Talon Metal Affinity Resin (Clontech, Mountain View, CA) (*Skau and Kovar, 2010*). Cdc8 was purified by boiling the cell lysate, performing an ammonium sulfate cut, and running on an anion exchange column (*Skau and Kovar, 2010*). His-tagged wild-type α-actinin Ain1 and mutant Ain1 (R216E) were expressed in High Five insect cells using baculovirus expression and purified using Talon Metal Affinity Resin (*Li et al., 2016*).

The $A_{280}$ of purified proteins was taken with a Nanodrop 2000c Spectrophotometer (Thermo-Scientific, Waltham, MA). Protein concentration was calculated using extinction coefficients Fim1: 55,140 $M^{-1}$ $cm^{-1}$, Cdc8 (WT and I76C mutant): 2,980 $M^{-1}$ $cm^{-1}$, Ain1 and Ain1(R216E): 86477 $M^{-1}$ $cm^{-1}$. Proteins were labeled with TMR-6-maleimide (Life Technologies, Grand Island, NY) or Cy5-monomaleimide (GE Healthcare, Little Chalfont, UK) dyes following manufacturer's protocols following purification. Proteins were flash-frozen in liquid nitrogen and stored at −80°C.

## Low-speed sedimentation assays

To perform low-speed sedimentation assays, 15 µM Mg-ATP actin monomers were spontaneously assembled in 10 mM imidazole, pH 7.0 50 mM KCl, 5 mM $MgCl_2$, 1 mM EGTA, 0.5 mM DTT, 0.2 mM ATP, and 90 µM $CaCl_2$ for 1 hr to generate F-actin. F-actin was then incubated with 500 nM Ain1 and a range of concentrations of Cdc8 (*Figure 6A–B*), or 5 µM Cdc8 and 0.5 µM Fim1-SNAP and/or 5 µM Ain1 (*Figure 7A–B*). This incubation occurred for 20 min at 25°C, followed by a 10,000 g spin for 20 min at room temperature. Supernatant and pellets were separated by 15% SDS-PAGE gel electrophoresis and stained for 30 min with Coomassie Blue, destained for 16 hr, and analyzed by densitometry with ImageJ.

## TIRF microscopy (TIRFM)

Time-lapse TIRFM movies were obtained using an Olympus IX-71 microscope with through-the-objective TIRF illumination, iXon EMCCD camera (Andor Technology), and a cellTIRF 4Line system (Olympus). The actin binding proteins (ABPs) of interest were initially added to a polymerization mix (10 mM imidazole (pH 7.0), 50 mM KCl, 1 mM $MgCl_2$, 1 mM EGTA, 50 mM DTT, 0.2 mM ATP, 50 µM $CaCl_2$, 15 mM glucose, 20 µg/mL catalase, 100 µg/mL glucose oxidase, and 0.5% (400 centipoise) methylcellulose). This ABP/polymerization mix was then added to Mg-ATP-actin (15% Alexa 488-labeled) to induce F-actin assembly in the presence of the ABPs of interest (*Zimmermann et al., 2016*). The mixture was then added to a flow chamber and imaged at room temperature at 5 s intervals (unless otherwise noted).

## Quantification of bundling

The percentage of actin filaments bundled was quantified at similar actin filament densities (between 2095 and 2295 µm total filament length) for each experiment. The total actin filament length in the chamber was measured manually by creating ROIs for every actin filament and measuring total actin filament length in FIJI (*Schindelin et al., 2012*; *Schneider et al., 2012*). ROIs for every segment of actin filament present in a bundle were then created and total bundled filament length measured. A 'bundled' segment was identified by looking at the history of the TIRF movie and determining that two actin filaments were associated and had been associated for at least three consecutive frames

(at least 15 s) including the frame quantified. The ratio of actin filament present in a bundle vs. total actin filament length was then calculated.

## Quantification of fluorescence intensity on actin filaments or bundles

Fluorescence intensity on actin filaments was quantified on movies taken under the same microscope conditions (laser intensity and angle, exposure time) and with the same protein batches. Fluorescence intensity was quantified at the same time point in each compared movie. The actin channel was used to identify single actin filaments or two-filament actin bundles, and ROIs of a three-pixel segmented line were created along all single filaments or two-filament bundles in the selected frame. The mean fluorescence for each segment was then measured using ImageJ.

## Quantifying number of cells with Ain1 in actin patches

To quantify the number of cells containing Ain1-GFP in actin patches, 1 min timelapse movies of 1 frame per second were taken, imaging both Ain1-GFP and an actin patch marker (ArpC5-mCherry or Fim1-mCherry). Movie files for independent experiments and replicates were blinded and independently analyzed for number of cells containing Ain1-GFP in actin patches using FIJI (*Schindelin et al., 2012*; *Schneider et al., 2012*). For a single cell to count as positively containing Ain1-GFP in actin patches, three criteria had to be met: 1) at least one distinguishable actin patch containing Ain1-GFP was observed, 2) the observed actin patch(es) contained Ain1-GFP for at least three frames and 3) the Ain1-GFP signal trajectory matched the channel expressing either ArpC5-mCherry or Fim1-mCherry. Total number of cells and cells with actin patches containing Ain1-GFP were then calculated to obtain percent of cells containing Ain1-GFP in actin patches.

## Quantification of Ain1 spot density

TIRFM images were collected with a cellTIRF 4Line system (Olympus) fitted to an Olympus IX-71 microscope with through-the-objective TIRF illumination and an iXon EMCCD camera (Andor Technology, Belfast, UK). 1.5 µM Mg-ATP-actin (10% Alexa 488 labeled) was mixed with polymerization TIRF buffer [10 mM imidazole (pH 7.0), 50 mM KCl, 1 mM $MgCl_2$, 1 mM EGTA, 50 mM DTT, 0.2 mM ATP, 50 µM $CaCl_2$, 15 mM glucose, 20 µg/mL catalase, 100 µg/mL glucose oxidase, and 0.5% (400 centipoise) methylcellulose] to induce F-actin assembly, 0.5 µM 0.5% TMR-labeled α-actinin, and 2 µM (monomer 1 µM dimer) unlabeled tropomyosin. This mixture was transferred to a flow cell for imaging at room temperature. Once the actin had polymerized and formed bundles, we imaged once in the 488 channel to visualize the labeled actin (one frame, 488 nm excitation for 50 ms) and then continuously imaged in the 561 channel to visualize the sparsely labeled α-actinin (100 frames, 561 nm excitation for 50 ms,~110 ms interval).

To measure α-actinin Ain1 spot density, we created kymographs on bundles for each experiment using ImageJ. Ain1 spots were detected in the kymograph as lines at least four pixels wide with a fluorescence value above 1.25 times that of background fluorescence. Spot density was normalized to the length of actin filaments in the bundle. Ain1 spot density was determined using the following formula:

$$\frac{n \div (L \times r)}{t}$$

where $n$ is the number of Ain1 spots detected, $L$ is the length of the bundle in µm, $r$ is the actin fluorescence ratio (total amount of fluorescence in the actin bundle divided by the average fluorescence of single actin filaments), and $t$ is the time of measurement in seconds.

## Acknowledgements

This work was supported by NIH R01 GM079265 and ACS RSG-11-126-01-CSM (to DRK), NIH MCB Training Grant T32 GM0071832 (to JRC, KEH, and MEO), Initiative for Maximizing Student Development (IMSD) NIGMS R25GM109439 (to MEO) and NSF Graduate Student Fellowships DGE-1144082 (to JRC), DGE-1144082 and DGE-1746045 (to AJH). Additional support was provided to DRK by the University of Chicago MRSEC, funded by the NSF through grant DMR-1420709. We thank Charlie

Dulberger and Yujie Li for assistance with Ain1 purification. We also thank Jonathan Winkelman and the Kovar lab for helpful discussions.

## Additional information

### Funding

| Funder | Grant reference number | Author |
|---|---|---|
| National Science Foundation | Graduate Research Fellowship DGE-1144082 | Jenna Christensen Alyssa J Harker |
| National Institutes of Health | T32 GM0071832 | Jenna Christensen Kaitlin E Homa Meghan O'Connell |
| National Science Foundation | Graduate Research Fellowship DGE-1746045 | Alyssa J Harker |
| National Institutes of Health | IMSD R25GM109439 | Meghan O'Connell |
| National Institutes of Health | R01 GM079265 | David R Kovar |
| American Cancer Society | RSG-11-126-01-CSM | David R Kovar |

The funders had no role in study design, data collection and interpretation, or the decision to submit the work for publication.

### Author contributions

Jenna R Christensen, Conceptualization, Data curation, Formal analysis, Investigation, Writing—original draft, Writing—review and editing; Kaitlin E Homa, Conceptualization, Formal analysis, Investigation, Writing—original draft, Writing—review and editing; Alisha N Morganthaler, Rachel R Brown, Cristian Suarez, Alyssa J Harker, Meghan E O'Connell, Formal analysis, Investigation; David R Kovar, Conceptualization, Supervision, Funding acquisition, Writing—review and editing

### Author ORCIDs

Jenna R Christensen (iD) https://orcid.org/0000-0003-0323-6169
David R Kovar (iD) https://orcid.org/0000-0002-5747-0949

### Decision letter and Author response

Decision letter https://doi.org/10.7554/eLife.47279.022
Author response https://doi.org/10.7554/eLife.47279.023

## Additional files

### Supplementary files

• Transparent reporting form
DOI: https://doi.org/10.7554/eLife.47279.020

### Data availability

All data generated or analyzed during the study are included in the manuscript.

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
