## [Decision Letter]

[Editors’ note: a previous version of this study was rejected after peer review, but the authors submitted for reconsideration. The first decision letter after peer review is shown below.]

Thank you for submitting your work entitled "Tropomyosin and a-actinin cooperation inhibits fimbrin association with actin filament networks in fission yeast" for consideration by *eLife*. Your article has been reviewed by three peer reviewers, including Mohan Balasubramanian as the Reviewing Editor, and the evaluation has been overseen by a Senior Editor. The reviewers have opted to remain anonymous.

Our decision has been reached after consultation between the reviewers. Based on these discussions and the individual reviews below, we regret to inform you that your work will not be considered further for publication in *eLife*.

The reviewers appreciate the novel approaches you are combining to address how distinct actin cross-linkers localize to distinct cellular locations and actin structures and how cdc8 and ain1 cooperate to exclude fimbrin from actomyosin rings and therefore themselves occupying the actomyosin ring. They find the conclusions to be of definite interest. However, it was felt that this paper did not reach the same standards set by the previous work. In light of this, we are unable to consider the work further presently, since the conclusions may change after the quantitations recommended by the reviewers.

However, if you feel you can address the issues raised satisfactorily through additional experiments we will be happy to consider this story as a new submission and will try to send it to the same three reviewers.

*Reviewer #1:*

Christensen et al. found that fission yeast α-actinin Ain1 competes with fimbrin Fim1 and tropomyosin Cdc8 is important for the competition. Using fission yeast genetics and single molecule experiments they make the conclusion that α-actinin and cdc8 outcompete fimbrin in vivo and in vitro. The results are quite robust and convincing in general and the in vitro competition studies with multi-color TIRF microscope are particularly interesting. However, I believe the manuscript needs significant revisions before further consideration in *eLife* Research Advance. Despite this work being an advance it is the feeling of this reviewer that the "advance" does not have the same high standards set by the previous work, in terms of quantitation and discussion.

1) The authors do not provide any explanation of how CK666 treatment causes Fim1 to move from cytoplasm to the AM ring while opposite effect is observed for Ain1.

2) The authors claim that Cdc8 enhances Ain1 activity 10 fold but provide only in vitro evidence with very questionable quantification approach. Moreover authors do not show where and how much of Ain1 is bound to F-actin or address why they can't label Ain1 directly. Finally, no biochemical evidences provided to confirm Cdc8 mediated increase of Ain1 bundling activity (e.g. pulldown assay).

3) Manuscript does not provide any information about the means of Cdc8 mediated augmentation of Ain1 bundling activity.

4) Figure 1 and later in vitro findings seem to contradict each other. On one hand CK666 treated cells display replacement of Ain1 with Fim1 in the ring while Cdc8 stays in the AM ring. On the other hand Fim1 fails to decorate Ain1/Cdc8 decorated F-actin bundles in TIRF experiments. Why is Fim1 able to displace Ain1 in vivo but not in vitro? Authors briefly speculate that increase of Fim1 concentration in the cytoplasm might be a factor but do not look into that in details.

5) Contractile ring fluorescence quantification in Fim1-GFP expressing cells

They quantified fluorescence of contractile rings but I could not understand how can they measure the ring fluorescence in Fim1-GFP cells treated with DMSO (Figure 1A). How can they differentiate the signals from the patches and rings? They can show how did they draw the ROI for this quantification.

6) Ain1-GFP patches: The results showing Ain1-GFP patches in fim1-1∆ cells (Figure 1F) are too preliminary to demonstrate their conclusions. Could they express Ain1-GFP more by using stronger promoters (e.g. Pnmt1, Padh1, Ptif51 and Pef1a)?

7) Expression levels of Ain1-GFP. The authors over-expressed Ain1-GFP and its R216E mutant in some experiments (Figure 1E-G and Figure 2C). Since they analysed localization of Ain1-GFP in different strains, they should show the expression levels of Ain1-GFP are comparable in these strains.

8) Cdc8-Cy5 and Cdc8-TMR: The author should use Cdc8-TMR for Figure 3E to demonstrate TMR did not inhibit the competition between Cdc8 and actin bundling proteins.

9) Affinity of Ain1 on F-actin filaments in the presence of Cdc8: The author should show the k_off_ values of Ain1 on F-actin in the presence of Cdc8 and Fim1 in the presence of Ain1 and Cdc8.

10) Can Ain1(R216E) release Fim1 from F-Actin in the absence of Cdc8?

The authors clearly demonstrate that Ain1 and Cdc8 outcompete Fim1 for F-actin binding in vitro (Figure 5). They can also test whether Ain1 R216E mutation can bypath the requirement of Cdc8 to outcompete Fim1 in vitro. If R216E can bypath the Cdc8 requirements, they can conclude that this out-competition between Ain1 and Fim1 is simply due to increased affinity of interaction between Ain1 and F-actin by Cdc8.

11) Genetic interaction between cdc8 and ain1: Related to 10. The author should test whether expression of Ain1 R216E can rescue cdc8 mutant or not.

*Reviewer #2:*

In this manuscript, Christensen et al., analyze the mechanisms regulating selective exclusion of Fim1 from the contractile ring. Using broad design principles of yeast genetics, live cell fluorescent imaging, and invitro TIRF microscopy, the authors propose that contractile ring component Cdc8 enhances bundling activity of Ain1 and together these actin binding components inhibit association of Fim1 to the ring.

Fim1 localizes to actin patches in cells. This localization is altered in presence of Arp2/3 inhibitor CK666, that disrupts actin patches, resulting in increase in the soluble pool of Fim1 which now localizes to the contractile ring. Fim1 recruitment to the ring displaces Ain1, which seems to become more diffused. This phenotype of Ain1 can be rescued in fim1-1Δ cells.

The invitro data in Figure 2, 3, 4 and 5 demonstrates a reduction in binding of Fim1 on single and 2-filament actin bundles in presence of Ain1 and Ain1 (R216E), suggesting competition between the two actin partners. Fim1 binding to Actin can be further lowered in presence of Cdc8. Based on this the authors conclude that Cdc8 and Ain1 cooperate to displace Fim1 from contractile ring.

This paper is a continuation of an earlier article from the same author. In all, the manuscript is well-written with appropriate conclusions. The quality of data is of very good. However, their main finding that Cdc8 displaces Fim1 has been discussed in detail in their earlier paper (as part of Figure 5). It would have made the manuscript more impactful if the authors had probed possibilities of an upstream adaptor at the actin patches that could be sequestering Fim1, and preventing its recruitment to the ring in addition to the synergistic inhibition by Ain1 and Cdc8. The paper needs more data if one needs to understand the mechanisms regulating Fim1's selective exclusion from the contractile ring.

Fim1 binding to F actin is very stable unlike Ain1 which is an order of magnitude weaker and highly dynamic. This would have implication in the contractile ring dynamics and rates of contraction in conditions (both genetic and pharmacologic) when Fim1 is localized to the contractile ring. The authors either show or discuss such consequence.

*Reviewer #3:*

In this manuscript, the Kovar lab follows up on their previous paper regarding how different actin-binding proteins are sorted to different actin networks in vivo through competition for actin filament binding. Here, they address how the very strong actin filament binding protein Fim1 (fimbrin) does not localize to acto-myosin cytokinetic rings, unlike the weaker binders Cdc8 (tropomyosin) and Ain1 (a-actinin). They show that Ain1 and Cdc8 do not compete with each other for binding to actin filaments, but they both compete with Fim1. Although Fim1 outcompetes each individual protein, the combination of Ain1 and Cdc8 outcompetes Fim1. A major strength of this work is showing this competition both in vitro and in cells. It should be noted that the concentrations used in vitro do not reflect the cellular concentrations of these proteins in most cases. On balance, I found the manuscript an important contribution to the field, and a nice extension of their previous publication. I have listed several comments below that might be addressed to strengthen some claims and data.

1) The authors propose that actin patches "sequester" Fim1 to prevent it from outcompeting Cdc8 and Ain1 at contractile rings. The measured cellular concentration of Fim1 is approximately 4 μM, of which 68% is free in the cytoplasm (PMID: 20587778). This suggests that actin patch disruption (e.g. Figure 1) does not free up a significant number of Fim1 molecules. Clearly, the presence of actin patches prevents Fim1 localization from the contractile ring, but strict competition for binding to actin filaments might be less important than other regulatory mechanisms.

2) Figure 5: the presence of Ain1 causes a 90% decrease in Fim1 levels on actin filaments, but only 16% increase Cdc8 levels on actin filaments. The difference in these two numbers leaves me wondering how much Ain1 is on these filaments, which has implications for the underlying mechanism. The authors could test the levels of filament-bound Ain1 in this experiment. On one hand, there might be lots of Ain1 that causes the drastic decrease in Fim1 due to simple competition. On the other hand, low levels of Ain1 (with Cdc8) might alter the structure of filaments and bundles in a manner that prevents Fim1 binding.

3) The authors use off-rates to explain competition with Ain1. Cdc8 and Ain1 cooperatively compete with Fim1, despite Ain1's fast off-rate for naked filaments. Does Ain1 have a slower off-rate for Cdc8-bound filaments?

---

## [Author Response]

[Editors’ note: the author responses to the first round of peer review follow.]

Reviewer #1:1) The authors do not provide any explanation of how CK666 treatment causes Fim1 to move from cytoplasm to the AM ring while opposite effect is observed for Ain1.

We have provided an explanation in the first Results section, which is the basis for the remainder of the study. Increasing the concentration of soluble Fim1 in the cell via CK666 treatment allows Fim1 to associate with the contractile ring and outcompete its contractile ring competitor Ain1.

2) The authors claim that Cdc8 enhances Ain1 activity 10 fold but provide only in vitro evidence with very questionable quantification approach. Moreover authors do not show where and how much of Ain1 is bound to F-actin or address why they can't label Ain1 directly. Finally, no biochemical evidences provided to confirm Cdc8 mediated increase of Ain1 bundling activity (e.g. pulldown assay).

We do not agree with the reviewer that TIRF microscopy imaging of Ain1-mediated F-actin bundling in the absence and presence of Cdc8 in vitro is a “very questionable quantification approach”, which we have used in previous studies (Li et al., 2016) and allows a direct visualization of bundling.

However, we do agree with the reviewer that using multiple approaches with further quantitative analysis and direct visualization of Ain1 would enhance the impact of our study. Therefore, we did everything suggested by the reviewer, as described in detail below, which has made the manuscript significantly more mechanistically insightful.

First, to provide biochemical evidence that Cdc8 increases the bundling efficiency of Ain1, we performed new low speed sedimentation assays that demonstrate how increasing concentrations of Cdc8 significantly enhance Ain1-mediated bundling (new Figure 6A, B). We also performed new low-speed sedimentation assays with Cdc8 and either Fim1, Ain1, or both (new Figure 7). The following text has been added to reflect these results: “Almost two-fold more Cdc8 pellets with Ain1-mediated F-actin bundles than with Fim1-mediated bundles (Figure 7B). Additionally, in the presence of both Fim1 and Ain1, an intermediate amount of Cdc8 pellets in the presence of both Fim1 and Ain1, suggesting that Ain1 allows Cdc8 to better associate with F-actin in the presence of Fim1.”

Second, we labeled Ain1 directly on cysteine residues and performed new 4-color TIRFM actin assembly assays to directly visualize the interplay between Ain1, Cdc8, and Fim1 (see new Figure 8). We added the following text describing these data: “To directly investigate the effect of Ain1 and Cdc8 cooperation on competition with Fim1, we performed four-color TIRFM with fluorescently labeled ABPs and quantified Fim1 association with F-actin in the presence of Cdc8 and/or Ain1. […] The synergy between Cdc8 and Ain1 may explain why Fim1 only poorly associates with contractile rings in fission yeast cells (Figure 8E).”

3) Manuscript does not provide any information about the means of Cdc8 mediated augmentation of Ain1 bundling activity.

In the Discussion section of the original submission, we proposed two possible mechanisms for Cdc8-mediated enhancement of Ain1 bundling activity: either (A) Cdc8 is changing the binding dynamics of Ain1 on F-actin bundles or (B) Cdc8 enhances the persistence length of F-actin, indirectly facilitating enhanced bundle formation.

To test model (A), we conducted new rapid TIRFM imaging experiments of sparsely-labeled Ain1 on uncoated versus Cdc8-coated F-actin (new Figure 6E-G). We discovered that “Three-fold more Ain1 binding events were observed on Cdc8-coated F-actin compared to uncoated F-actin (Figure 6E-G), suggesting that Cdc8 enhances the binding of Ain1 to F-actin, thereby increasing the F-actin bundling ability of Ain1.”

We cannot rule out that the persistence length of F-actin also plays a role in the increased Ain1-mediated bundling seen in the presence of Cdc8. Therefore, we also added the following text to the Discussion: “We suspect that conformational changes in the actin filament may also affect the observed increase of αactinin Ain1-mediated bundling in the presence of tropomyosin Cdc8 (Figure 6A-D). […] Tropomyosin’s increase of the persistence length of F-actin may have an additional effect on F-actin bundling by promoting bundle stability and inter-filament contacts.”

4) Figure 1 and later in vitro findings seem to contradict each other. On one hand CK666 treated cells display replacement of Ain1 with Fim1 in the ring while Cdc8 stays in the AM ring. On the other hand Fim1 fails to decorate Ain1/Cdc8 decorated F-actin bundles in TIRF experiments. Why is Fim1 able to displace Ain1 in vivo but not in vitro? Authors briefly speculate that increase of Fim1 concentration in the cytoplasm might be a factor but do not look into that in details.

Our new 4-color TIRFM actin assembly assays, suggested by Review #1, reveal that Fim1 and Ain1 do compete for binding F-actin in vitro (new Figure 8), consistent with in vivo results. The kymograph in Figure 8B (bottom panel) shows that Ain1 and Fim1 associating with F-actin bundles is mutually exclusively as they sort to non-overlapping domains on F-actin bundles. This is consistent with our in vivo results in Figure 1 and Figure 1—figure supplement 3, as Ain1 is still present on the ring after CK-666 treatment, but in significantly lower amounts.

5) Contractile ring fluorescence quantification in Fim1-GFP expressing cellsThey quantified fluorescence of contractile rings but I could not understand how can they measure the ring fluorescence in Fim1-GFP cells treated with DMSO (Figure 1A). How can they differentiate the signals from the patches and rings? They can show how did they draw the ROI for this quantification.

We agree with the reviewer that the predominant association of Fim1 with actin patches near the ring makes quantification of ring fluorescence difficult. This is precisely why we had also looked at different contractile ring stages (Figure 1—figure supplement 3), as there are fewer actin patches at the contractile ring site in Stage 1 and 2 cells. We have added a yellow box to Figure 1B to highlight the region measured for contractile ring fluorescence intensity in Fim1GFP cells. We also added the following text: “At all stages of contractile ring assembly and constriction, Fim1 similarly localizes to the contractile ring and displaces Ain1 following CK-666 treatment (Figure 1—figure supplement 3), though it is most prominent in stages with fully-developed contractile rings (stages 2 and 3).” Furthermore, if anything, the increase of Fim1 ring fluorescence in the presence of CK-666 is underreported given that a significant amount of Fim1 ring fluorescence in the absence of CK-666 is likely due to actin patches at the division site.

We also provided an explanation of quantification in the “Contractile ring fluorescence quantification” section of the Materials and methods: “Contractile ring maturation was divided into three stages by measuring the distance between spindle pole bodies (SPBs, visualized by Sad1-tdTomato) and noting constriction of the contractile ring. […] Maximum project were used for images in figures and sum projections were used for quantification.”

6) Ain1-GFP patches: The results showing Ain1-GFP patches in fim1-1∆ cells (Figure 1F) are too preliminary to demonstrate their conclusions. Could they express Ain1-GFP more by using stronger promoters (e.g. Pnmt1, Padh1, Ptif51 and Pef1a)?

We disagree that the results showing 41X-Ain1-GFP in actin patches (Figure 2 of the revised manuscript) are not conclusive, since Ain1-GFP is clearly observed to colocalize with ArpC5mCherry over time. For clarity, we added the following: “To increase the expression of Ain1-GFP, we introduced an additional copy of Ain1-GFP at the leu1-32 locus under the medium-strength 41Xnmt promoter (Li et al., 2016).” Furthermore, consistent with these findings, the less-dynamic mutant 41X-Ain1(R216E) localizes to actin patches even more strongly (Figure 4).

7) Expression levels of Ain1-GFP. The authors over-expressed Ain1-GFP and its R216E mutant in some experiments (Figure 1E-G and Figure 2C). Since they analysed localization of Ain1-GFP in different strains, they should show the expression levels of Ain1-GFP are comparable in these strains.

We appreciate this important comment from the reviewer. We performed an analysis of Ain1 expression in all strain backgrounds used (new Figure 2—figure supplement 1) and added the following text: “We observed that while most Ain1-GFP constructs are expressed similarly to endogenously-tagged Ain1, the amount of Ain1-GFP overexpressed under the 41Xnmt promoter is almost two-fold higher (Figure 2—figure supplement 1). […] Therefore, Fim1 and Ain1 appear to compete for association with F-actin at both actin patches and the contractile ring.”

8) Cdc8-Cy5 and Cdc8-TMR: The author should use Cdc8-TMR for Figure 3E to demonstrate TMR did not inhibit the competition between Cdc8 and actin bundling proteins.

Throughout the course of this study and previous work (Christensen et al. 2017), we have found that the specific identity of the fluorophore (TMR vs. Cy5) does not change the behavior or binding dynamics of Cdc8 on F-actin. Figure 5C (formerly Figure 3E) serves as a control to show that Fim1 and Cdc8 compete for binding to F-actin, as we have shown previously (the subject of Christensen et al., 2017). The true takeaway from Figure 5 is that neither Ain1 (Figure 5A) nor Ain1(R216E) (Figure 5B) displace Cdc8. That Ain1 and Cdc8 cooperate, whereas Fim1 and Cdc8 compete for associating with F-actin, is also observed in new Figure 7 and new Figure 8.

9) Affinity of Ain1 on F-actin filaments in the presence of Cdc8: The author should show the k_off_ values of Ain1 on F-actin in the presence of Cdc8 and Fim1 in the presence of Ain1 and Cdc8.

We appreciate the reviewer’s suggestions, and have therefore determined whether the F-actin binding dynamics of Ain1 were altered in the presence of Cdc8 with new experiments. We conducted rapid TIRFM of sparsely-labeled Ain1 on uncoated versus Cdc8-coated F-actin (new Figure 6E-G). We found that “Three-fold more Ain1 binding events were observed on Cdc8 coated F-actin compared to uncoated F-actin (Figure 6E-G), suggesting that Cdc8 enhances the binding of Ain1 to F-actin, thereby increasing the F-actin bundling ability of Ain1.” However, we were not able to determine whether the 3-fold increase in observed Ain1 association events in the presence of Cdc8 are due to an increase in the rate of Ain1 binding and/or a decrease in the rate of Ain1 dissociation.

10) Can Ain1(R216E) release Fim1 from F-Actin in the absence of Cdc8?The authors clearly demonstrate that Ain1 and Cdc8 outcompete Fim1 for F-actin binding in vitro (Figure 5). They can also test whether Ain1 R216E mutation can bypath the requirement of Cdc8 to outcompete Fim1 in vitro. If R216E can bypath the Cdc8 requirements, they can conclude that this out-competition between Ain1 and Fim1 is simply due to increased affinity of interaction between Ain1 and F-actin by Cdc8.

In Figure 3 (was Figure 2), we show that Ain1R216E does not outcompete Fim1 more than WT Ain1.

11) Genetic interaction between cdc8 and ain1: Related to 10. The author should test whether expression of Ain1 R216E can rescue cdc8 mutant or not.

We did not do the suggested genetic experiment as we think this is very unlikely as Cdc8 is involved with regulating the function of several other essential ABPs, such as myosin-2 Myo2.

Reviewer #2:[…] This paper is a continuation of an earlier article from the same author. In all, the manuscript is well-written with appropriate conclusions. The quality of data is of very good. However, their main finding that Cdc8 displaces Fim1 has been discussed in detail in their earlier paper (as part of Figure 5). It would have made the manuscript more impactful if the authors had probed possibilities of an upstream adaptor at the actin patches that could be sequestering Fim1, and preventing its recruitment to the ring in addition to the synergistic inhibition by Ain1 and Cdc8. The paper needs more data if one needs to understand the mechanisms regulating Fim1's selective exclusion from the contractile ring.Fim1 binding to F actin is very stable unlike Ain1 which is an order of magnitude weaker and highly dynamic. This would have implication in the contractile ring dynamics and rates of contraction in conditions (both genetic and pharmacologic) when Fim1 is localized to the contractile ring. The authors either show or discuss such consequence.

We thank the reviewer for the favorable comments. However, we disagree that the main point of this manuscript “has been discussed in detail in their earlier paper…”. The main points of this important new manuscript are that although Fim1 outcompetes Cdc8 in isolation (the main point of our earlier *eLife* paper), (1) the association of fimbrin Fim1 with actin patches acts as a sink that contributes to preventing Fim1 from associating with contractile rings more strongly, (2) the contractile ring ABP a-actinin Ain1 is another important player in ABP sorting, (3) fimbrin Fim1 also outcompetes Ain1 for binding actin filaments in isolation, like it does tropomyosin Cdc8, but (4) tropomyosin Cdc8 enhances the F-actin bundling activity of a-actinin Ain1, and (5) the combination of tropomyosin Cdc8 and a-actinin Ain1 cooperate to successfully compete with fimbrin Fim1 for binding actin filaments, further explaining why fimbrin Fim1 does not associate more strongly with contractile rings in vivo.

Understanding how Fim1 might be sequestered in actin patches or inactivated by posttranslational modifications is an ongoing topic of study in our lab, but outside the scope of the current manuscript. We suspect that several mechanisms, including post-translational modifications, actin filament conformation, and the dynamics of ABPs, contribute to Fim1’s preferential association with actin patches and affect the competitive and cooperative interactions between ABPs.

Regarding the dynamics of bundling proteins, we thank the reviewer for highlighting this important point and have added the following new passage to the Discussion: “We discovered that the contractile ring ABP α-actinin Ain1 competes with fimbrin Fim1, and that dynamic association of both Ain1 and Fim1 with actin filaments affects their ability to compete at different F-actin networks. […] Though our results suggest that Ain1 is more dynamic on uncoated actin filaments, we showed that wild-type Ain1 is still a dynamic bundler on Cdc8-coated actin filaments and is likely capable of allowing contractile ring ABPs to remain associated with and function optimally at the contractile ring.”

Reviewer #3:[…] [1) The authors propose that actin patches "sequester" Fim1 to prevent it from outcompeting Cdc8 and Ain1 at contractile rings. The measured cellular concentration of Fim1 is approximately 4 μM, of which 68% is free in the cytoplasm (PMID: 20587778). This suggests that actin patch disruption (e.g. Figure 1) does not free up a significant number of Fim1 molecules. Clearly, the presence of actin patches prevents Fim1 localization from the contractile ring, but strict competition for binding to actin filaments might be less important than other regulatory mechanisms.

We definitely agree with the reviewer about the importance of additional regulatory mechanisms, and we have updated the Discussion to reflect this. It is currently unknown how much of the free pool of Fim1 is active. We have added a section discussing how posttranslational modification of Fim1 could also contribute to regulation of its activity: “in vitro, the combined efforts of α-actinin Ain1 and tropomyosin Cdc8 prevent ~40% of fimbrin Fim1 association with F-actin bundles. […] Fission yeast Fim1 might be similarly post-translationally modified (Kettenbach et al., 2015; Swaffer et al., 2018), and therefore a portion of the cytoplasmic Fim1 pool might be more or less active.”

2) Figure 5: the presence of Ain1 causes a 90% decrease in Fim1 levels on actin filaments, but only 16% increase Cdc8 levels on actin filaments. The difference in these two numbers leaves me wondering how much Ain1 is on these filaments, which has implications for the underlying mechanism. The authors could test the levels of filament-bound Ain1 in this experiment. On one hand, there might be lots of Ain1 that causes the drastic decrease in Fim1 due to simple competition. On the other hand, low levels of Ain1 (with Cdc8) might alter the structure of filaments and bundles in a manner that prevents Fim1 binding.

We agree completely with the reviewer that direct visualization of Ain1 would greatly increase the impact of our study. Therefore, we labeled Ain1 directly on cysteine residues and performed new 4-color TIRFM actin assembly assays to directly visualize the interplay between Ain1, Cdc8, and Fim1 (see new Figure 8). We added the following text describing these data: “To directly investigate the effect of Ain1 and Cdc8 cooperation on competition with Fim1, we performed four-color TIRFM with fluorescently labeled ABPs and quantified Fim1 association with F-actin in the presence of Cdc8 and/or Ain1. […] The synergy between Cdc8 and Ain1 may explain why Fim1 only poorly associates with contractile rings in fission yeast cells (Figure 8E).”

3) The authors use off-rates to explain competition with Ain1. Cdc8 and Ain1 cooperatively compete with Fim1, despite Ain1's fast off-rate for naked filaments. Does Ain1 have a slower off-rate for Cdc8-bound filaments?

Please see our response to reviewer 1, comment 9.